# Fast, Direct Dihydrouracil Quantitation in Human Saliva: Method Development, Validation, and Application

**DOI:** 10.3390/ijerph19106033

**Published:** 2022-05-16

**Authors:** Beatrice Campanella, Tommaso Lomonaco, Edoardo Benedetti, Massimo Onor, Riccardo Nieri, Federica Marmorino, Chiara Cremolini, Emilia Bramanti

**Affiliations:** 1National Research Council of Italy, C.N.R., Institute of Chemistry of Organometallic Compounds—ICCOM, Via G. Moruzzi 1, 56124 Pisa, Italy; beatrice.campanella@pi.iccom.cnr.it (B.C.); onor@pi.iccom.cnr.it (M.O.); riccardonieri.nutrizionista@gmail.com (R.N.); 2Department of Chemistry and Industrial Chemistry, University of Pisa, Via G. Moruzzi 15, 56124 Pisa, Italy; tommaso.lomonaco@unipi.it; 3Hematology Unit, Department of Oncology, Azienda Ospedaliero Universitaria Pisana, Via Roma 67, 56127 Pisa, Italy; edobenedetti@gmail.com; 4Unity of Oncology, Department of Translational Research and New Technologies in Medicine, University of Pisa, Via Roma 67, 56127 Pisa, Italy; federica.marmorino@gmail.com (F.M.); chiaracremolini@gmail.com (C.C.)

**Keywords:** saliva, dihydrouracil, chemotherapy metabolite, high-performance liquid chromatography

## Abstract

*Background*. Salivary metabolomics is garnering increasing attention in the health field because of easy, minimally invasive saliva sampling. Dihydrouracil (DHU) is a metabolite of pyrimidine metabolism present in urine, plasma, and saliva and of fluoropyrimidines-based chemotherapeutics. Its fast quantification would help in the identification of patients with higher risk of fluoropyrimidine-induced toxicity and inborn errors of pyrimidine metabolism. Few studies consider DHU as the main salivary metabolite, but reports of its concentration levels in saliva are scarce. We propose the direct determination of DHU in saliva by reversed-phase high-performance liquid chromatography (RP-HPLC-UV detector) as a simple, rapid procedure for non-invasive screening. *Methods*. The method used was validated and applied to 176 saliva samples collected from 21 nominally healthy volunteers and 4 saliva samples from metastatic colorectal cancer patients before and after receiving 5-fluorouracil chemotherapy. *Results.* DHU levels in all samples analyzed were in the μmol L^−1^ range or below proving that DHU is not the main metabolite in saliva and confirming the results found in the literature with LC-MS/MS instrumentation. Any increase of DHU due to metabolism dysfunctions can be suggestive of disease and easily monitored in saliva using common, low-cost instrumentation available also for population screening.

## 1. Introduction

Dihydrouracil (DHU) is a small metabolite of interest in clinical chemistry because, with dihydrothymine, it is the by-product of the pyrimidine metabolism [1]. Its increase in biological fluids is often a symptom of a disfunction of dihydropyrimidinase (DHP), the enzyme involved in the pyrimidine base degradation (uracil and thymine). DHP deficiency gives inborn errors of pyrimidine metabolism, with symptoms ranging from asymptomatic cases to epilepsy, intellectual disability, epilepsy, and autism [1,2,3]. 5-fluorouracil and its oral pre-prodrug capecitabine are the election treatment of colorectal, pancreatic, gastric, breast, and head and neck cancers [4]. The fluoropyrimidine toxicity (infections, septicemia, leucopenia, thrombocytopenia, neutropenia, angina pectoris, myocardial infarction, and gastrointestinal adverse effects) is due to an inherited deficiency of DHP in 15% of patients, which is responsible for 80% of the catabolism of fluoropyrimidines [5]. This limits or delays the administration of optimal or successive courses in chemotherapy treatments [6]. The detection of DPD activity is indeed recommended by the European Medicine Agency (https://www.ema.europa.eu/en/news/ema-recommendations-dpd-testing-prior-treatment-fluorouracil-capecitabine-tegafur-flucytosine) (accessed on 13 May 2022) before starting any treatment with fluoropyrimidine [7,8]. Thus, the quick and easy determination of DHU and of DHU-to-uracil (U) concentration ratio (DHU/U) in biological fluids (e.g., plasma, urine, and saliva) is a significant analytical target due to the potential relationship between these compounds and DHP deficiency.

Many analytical techniques have been developed for the determination of DHU in plasma based on UV [9,10,11,12,13] and mass spectrometry (MS) detection, including a fully automated LC-MS/MS method for the accurate and robust quantification of U and DHU in plasma [5,6]. They found that DHU in plasma ranges between 0.002–0.011 μmol L^−1^. Pan et al. recently reported an improved method for the detection of uracil and dihydrouracil in human plasma using reversed-phase high-performance liquid chromatography. Tsuchiya et al. determined DHU in urine by GC-MS and LC-MS reporting reference values in Fujita Health University Hospital in the range of 0–16 μmol mmol^−1^ creatine (corresponding to 0–0.416 μmol L^−1^ considering the reference interval of creatinine) [1]. Van Kuilenburg et al. quantified DHU in urine, plasma, and cerebrospinal fluid using reversed-phase LC combined with electrospray MS/MS [14], finding values in agreement with those reported above. Sparidans et al. reported a concentration of DHU in urine of 1.1 ± 0.9 μmol L^−1^ in healthy subjects determined by LC–MS/MS [15]. Analogous values were obtained by Sun et al. in urine by LC-MS [16]. Few data have been reported on the quantitation of DHU in saliva, which is an attractive matrix due to its non-invasive sampling. Venzon Antunes et al. validated a LC-MS/MS assay for the measurement of U and DHU concentrations in dried saliva spots of 77 individuals (38 healthy volunteers and 39 patients with gastrointestinal cancer scheduled to receive 5-fluorouracil chemotherapy) for the evaluation of DHP enzyme activity, finding a median of 0.926 μmol L^−1^ (0.673–1.798 μmol L^−1^ range) [17,18]. Galarza et al. measured by LC–MS/MS endogenous plasma and salivary DHU in 60 patients with gastrointestinal malignancies [19]. They found that in plasma DHU had a median of 0.799 μmol L^−1^ (range 0.537–1.023 μmol L^−1^); in saliva, DHU had a median of 2.168 μmol L^−1^ (range 1.139–5.013 μmol L^−1^) [19]. Al-Shehri et al. found DHU concentration level < 3 μmol L^−1^ in saliva of 10 healthy adults and 8.6 μmol L^−1^ in 10 neonates [3]. Carlsson et al. determined increasing levels of DHU in saliva of 73 colorectal cancer patients treated with 5-fluorouracil-based chemotherapy by HPLC method with UV detector at 220 nm using a first C18 reversed-phase column and by switching the C18 reversed-phase column to a second cation-exchange column for one minute to separate uracil from DHU [20]. Only after chemotherapy, they found DHU concentration of 0.043 ± 0.035 μmol L^−1^. In contrast with these values found in saliva, other fundamental papers on salivary metabolomics [21,22] reported DHU being the main metabolite of saliva itself, about 2000 μmol L^−1^, i.e., a concentration 1000 higher than that found in saliva, plasma, and urine by the previously mentioned works. DHU concentration has been reported to be 2168 ± 128 μmol L^−1^ by Dame et al. [22] and 2210 ± 353 μmol L^−1^ by Sugimoto et al. [21], measured by HPLC–UV and capillary electrophoresis-MS, respectively.

We recently approached the study of salivary metabolites with the aim of using saliva analysis as a minimally invasive, safe, and painless tool for the monitoring of the health status [23,24,25,26]. Here, we propose a fast bioanalytical procedure for the determination DHU in saliva based on saliva dilution and analysis reversed-phase liquid chromatography with diode array detection (RP-LC-DAD) in less than 6 min isocratic separation. No interferences from endogenous components were revealed. The method was validated and applied to 176 saliva samples collected from 21 nominally healthy volunteers and 4 saliva samples from metastatic colorectal cancer patients before and after receiving maintenance therapy with 5-fluorouracil chemotherapy.

## 2. Materials and Methods

### 2.1. Reagents, Materials, and Reference Standard Samples

Dihydrouracil (D-7628, CAS n. 504-07-4) was purchased from Sigma Aldrich (Milan, Italy). Methanol HPLC grade was purchased from Merck (Darmstadt, Germany). Purified water was obtained from an Elga Purelab Ultra system from Veolia Labwater (High Wycombe, United Kingdom). Additionally, 0.20 μm RC Mini-Uniprep filter units were obtained from (Agilent Tech., Milan, Italy). Saliva samples were collected using a Salivette device (Sarstedt, Nümbrecht, Germany).

### 2.2. Preparation of Solutions and Standards

DHU stock solution was prepared in water at the concentration of 10 mmol L^−^^1^. Working solutions (concentrations 1–2500 μmol L^−^^1^) were prepared by dilution of the stock solutions with the eluent phase (5 mM sulfuric acid).

### 2.3. Saliva Sample Preparation

Salivette^®^ roll-shaped polyester swabs (Sarstedt, Nümbrecht, Germany) were used for the collection of non-stimulated saliva samples collected in different days from 21 nominally healthy volunteers and from 4 metastatic colorectal cancer patients receiving 5-fluorouracil at the University Hospital of Pisa (Pisa, Italy) within 1 h from the beginning of the 5-fluorouracil infusion and at the end of the infusion. 5-fluorouracil was administered as a continuous 48 h intravenous infusion at a dose of 2400 mg/mq (maintenance therapy). In total, *N* = 176 saliva samples were collected from healthy volunteers sampling in different days. This study was conducted in accordance with the principles set in the Helsinki Declaration. Ethical review and approval are not applicable because all subjects were volunteers. All subjects provided their written informed consent before any procedure. The healthy participant population included 11 men and 10 women, ranging in age from 26 to 62 y (mean age ± standard deviation, 40 ± 12 y); patients included 1 man and 3 women, ranging in age from 52 to 78 y (64 ± 14 y). For all participants, saliva samples were collected at the same time of the day (6:00 a.m. to 7:00 a.m.), to avoid fluctuation in the results due to the circadian saliva cycle, after at least 8 h of fasting or tooth brushing. Salivette^®^ swabs (Sarstedt, Nümbrecht, Germany) were kept in the mouth for 5 min and, after collection were immediately stored at −20 °C. Prior to analysis, swabs were thawed at room temperature and then centrifuged at 4500× *g* for 10 min at 4 °C (Eppendorf™ 5804R Centrifuge) (Eppendorf, Milan, Italy). Saliva samples were diluted 5 times in 5 mM sulfuric acid, filtered using a 0.20 μm RC Mini-Uniprep filter, and then injected (V_inj_ = 5 μL) in the HPLC system.

### 2.4. HPLC Analytical Conditions

An Agilent 1260 Infinity HPLC system (G1311B quaternary pump) equipped with 1260 Infinity High-Performance Degasser, a TCC G1316A thermostat, 1260ALS autosampler (G1329B), and UV/vis diode array (1260 DAD G4212B) (Eppendorf, Milan, Italy) was employed. The identification of metabolites was based on the comparison of the retention time and UV spectra of standard compounds. The chromatographic separation was carried out by Zorbax Phenyl-Hexyl RP (Agilent Tech., Milan, Italy) 250 × 4.6 mm (silica particle size 5 μm) at 45 °C using an isocratic elution with 100% 5 mM sulfuric acid (pH 2.2) for 6 min. The column was rinsed with 100% methanol for 15 min, and then, a re-equilibration step was performed. All the solutions were filtered using a 0.22 μm regenerate cellulose filter (Millipore, Milan, Italy).

## 3. Results and Discussion

### 3.1. Method Validation

Figure 1A shows the absorbance chromatogram at 220 nm of 5, 10, and 45 μmol L^−^^1^ DHU eluting in our operating conditions at t_R_ = 5.151. The method was validated in saliva by evaluating linearity, limit of detection (LOD), limit of quantification (LOQ), accuracy, and precision. LOD and LOQ were determined by the analysis of samples with known concentrations of the standard analyte and by establishing the minimum level at which the analyte can be reliably detected and quantified with acceptable accuracy and precision, respectively. Linearity was calculated in the range 1–2500 μmol L^−^^1^ for the chemical standard DHU. Precision was recorded in standard solutions at three concentration levels in (QC_Low_ = 10 μmol L^−^^1^, QC_Medium_ = 200 μmol L^−^^1^, and QC_High_ = 1000 μmol L^−^^1^) in three replicates to calculate the percentage of relative standard deviation (RSD), resulting as 3.4, 2.1, and 1.1%, respectively. In saliva at QC_Low_ = 10 μmol L^−^^1^, QC_Medium_ = 500 μmol L^−^^1^, and QC_High_ = 2000 μmol L^−^^1^ RSD resulted as 6.1, 2.3, and 2.5%, respectively. DHU concentration was calculated based on the area of the peak at t_R_ = 5.151 min and the corresponding calibration curve obtained in the matrix.

The accuracy of the method was evaluated by spiking a pooled saliva sample with the DHU standard solution, diluting five times in 5 mM sulfuric acid, filtering using a 0.20 μm RC Mini-Uniprep filter, and then injecting (V_inj_ = 5 μL) in the HPLC system (*N* = 3 replicates). Figure 1B shows the absorbance chromatogram at 220 nm of unspiked saliva (continuous line) and spiked with 10 μmol L^−1^ DHU. The spike of DHU to saliva sample showed a peak eluting at t_R_ = 5.151 min, which was not detected in unspiked saliva. Figure 1C shows a detail of the absorbance chromatogram at 220 nm of unspiked saliva and saliva and spiked with 1, 3, 10, or 30 μmol L^−1^ DHU. Figure 1D shows the comparison of the external calibration curve (open circle) and the calibration curve of the analytical standards added in saliva (full circle). We also verified that DHU is quantitatively recovered (98 ± 3%) from the swab by analyzing 50 μmol L^−^^1^ DHU standard solution before and after adsorption/centrifugation of Salivette^®^. Table 1 reports the fitting parameters of the calibration curves (Figure 1D) and the quantitation limits (LOQs). Although the nominal LOQ of DHU in standard solutions is 0.103 μmol L^−^^1^, in saliva, LOQ spans from 0.103 to 3 μmol L^−^^1^ depending on the concentration level of α-ketoglutaric acid (shoulder at 5.022 min in Figure 1B), which has a variable concentration in saliva [26]. Figure 1B shows the worst situation, in which α-ketoglutaric acid has high concentration and coelutes with acetic acid.

### 3.2. Application

In all saliva samples of 18 out of 21 nominally healthy volunteers (total 161 saliva sample analyses), DHU was below the LOQ of this method. In three subjects, DHU concentration was found 4.01 ± 1.15 (nominally S19, *N* = 8 sampling in different days), 0.66 ± 0.67 (nominally S20, *N* = 3 sampling in different days), and 0.19 ± 0.14 μmol L^−1^ (nominally S21, *N* = 4 sampling in different days). Figure 2 shows representative absorbance chromatograms at 220 nm of saliva samples of S20 and S21 compared with the chromatogram of a nominally healthy volunteer, with DHU < LOQ (S18).

DHU concentration in saliva sample from four metastatic colorectal cancer patients receiving 5-fluorouracil sampled before and after the infusion was always below the LOQ. These results obtained agree with the DHU concentration value found in saliva by other authors [17,18,19,20], clarifying that DHU is not the main metabolite in saliva [21,22].

Table 2 summarizes the scarce data reported in the literature on DHU concentration in saliva, including the results found in this work.

Despite the limited number of cases examined, the finding of a slightly higher concentration of DHU in 3 out of 21 nominally healthy volunteers arouses interest in the role of DHU in metabolism. DHU monitoring is indeed commonly related to DPD dysfunctions during chemotherapy treatments. However, DHU is an oxidation product of the nucleotide uracil and may provide a stable marker of altered nucleotide metabolism in several diseases [14,28,29,30,31,32]. In inflammatory colon tissue (ulcerative colitis or Crohn’s disease), DPD activity was significantly higher than in normal tissue (*p* = 0.006) [33]. Wikoff et al. found distinct metabolic perturbations associated with early stage lung adenocarcinoma [34]. DHU was significantly elevated by 2.4-fold in cancer compared with control tissue and constituted the single best multivariate predictor for cancer [34]. Thus, the wide screening of DHU in an “easy” matrix such as saliva and also in not-highly equipped laboratories as well as eventually the definition of its reference range within a population is of the utmost importance for metabolic studies. This topic is beyond the methodological goal of this work, but in perspective, the screening of DHU in saliva could address the clinician to more informative, detailed investigations, which is the goal of personalized medicine.

## 4. Conclusions

The direct RP-HPLC-DAD method proposed allows the analyst to determine DHU in saliva in less than 6 min after an easy, fast dilution and filtration of the saliva sample. The DHU concentration levels found in saliva in this work as well as the values found by other authors [17,18,19,20] confirm that DHU is not the main metabolite in human saliva, and its concentration in nominally healthy volunteers is in the micromolar range or below. Despite many of these data have been obtained using LC-MS/MS approaches, we found that HPLC with UV detector can be a straightforward approach suitable for a fast, low-cost screening of DHU in saliva whenever the operating conditions allow the control of potential compounds interfering with the DHU determination. In HPLC, methods with UV detection interferences must indeed be accurately checked because of the scarce selectivity of UV spectra. The LOQ of DHU found using this methodology is indeed comparable with the LOD found in saliva by other authors using LC-MS/MS [3]. Due to the role of DHU as a marker of pyrimidine metabolism, the wide screening of DHU in a nominally healthy population may provide a personalized diagnostic to identify cohorts of patients with alterations in DPD activity.

## Figures and Tables

**Figure 1 ijerph-19-06033-f001:**
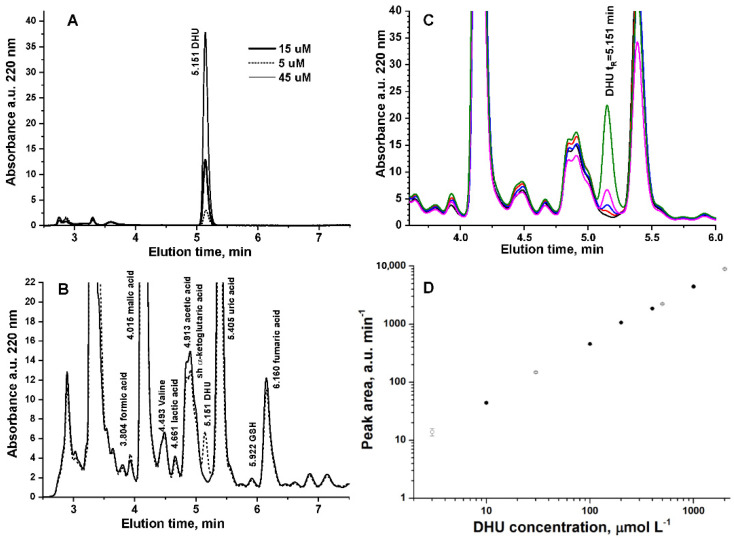
(**A**) Absorbance chromatogram at 220 nm of 5, 10, and 45 μmol L^−1^ DHU. (**B**) Absorbance chromatogram at 220 nm of unspiked saliva (continuous line) and spiked with 10 μmol L^−1^ DHU (dotted line). (**C**) Absorbance chromatogram at 220 nm of unspiked saliva (continuous line) and spiked with 1 (red line), 3 (blue line), 10 (magenta line), or 30 μmol L^−1^ DHU (green line). (**D**) External calibration curve (full circle) and calibration curve of DHU analytical standard added in saliva (open circle) (*N* = 3 replicates).

**Figure 2 ijerph-19-06033-f002:**
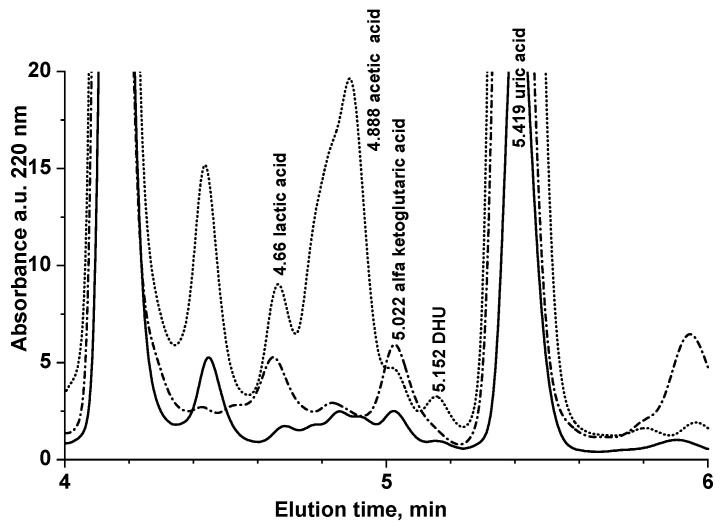
Representative absorbance chromatogram at 220 nm of saliva samples of S20 (dotted line, 0.66 ± 0.67 μmol L^−^^1^ DHU, *N* = 3 sampling in different days) and S21 (continuous line, 0.19 ± 0.14 μmol L^−^^1^ DHU, *N* = 4 sampling in different days) compared with the chromatogram of a nominally healthy volunteer with DHU < LOQ (S18, dash-dotted line).

**Table 1 ijerph-19-06033-t001:** Calibration curves and limit of quantitation LOQ for the determination of dihydrouracil DHU by HPLC-DAD (*N* = 3 replicates).

	External Calibration Curve	Analytical Standard Addition to Saliva
Slope (μmol^−1^ L)	4.443 ± 0.053	4.450 ± 0.044
Intercept	60.8 ± 26.18	−4.238 ± 50.9
R^2^	0.9981	0.9990
LOQ (μmol L^−1^)	0.103	0.103–3.0

The matrix effect was ruled out by the negligible difference (*p* > 0.05) observed between the external and internal calibration slope [27].

**Table 2 ijerph-19-06033-t002:** Summary of data on DHU quantitation in saliva.

Data Set	Method	Concentration (μmol L^−1^)	Ref.
73 colorectal cancer patients treated with 5-fluorouracil-based chemotherapy	HPLC method (Reversed Phase and cation exchange)	0.043 ± 0.035 (after chemotherapy)	[20]
10 healthy adults 10 neonates		<3 in adults 8.6 in neonates	[3]
38 healthy volunteers 39 patients (gastrointestinal cancer treated with 5-fluorouracil chemotherapy)	LC-MS/MS (dried spot saliva)	0.926 (median) (range 0.673–1.798)	[17,18]
60 patients with gastrointestinal malignancies	LC–MS/MS	2.168 (median) (range 1.139–5.013)	[19]
155 healthy adult volunteers	Capillary electrophoresis-MS	2210 ± 353	[21]
16 healthy adult volunteers	HPLC–UV	2168 ± 128	[22]
21 healthy volunteers + 4 colorectal cancer patients treated with 5-fluorouracil-based chemotherapy (176 saliva samples collected in different days)	RP-HPLC-UV	*N* = 18 + 4 subjects < LOQ ^1^ *N* = 3 healthy subjects 0.19–4.01 (range)	This work

^1^ LOQ = 0.103–3 μmol L^−1^.

## Data Availability

Not applicable.

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
