# Peer review of "Fast, Direct Dihydrouracil Quantitation in Human Saliva: Method Development, Validation, and Application"

_ijerph, 2022, doi:10.3390/ijerph19106033_

Round 1
Reviewer 1 Report
- In line 17, the clarity of the presentation needs to be improved by providing a short introduction to HPLC-DAD.
- In line 67, the presentation of LC-MS/MS endogenous plasma and salivary DHU and U/DHU ratios may need to be rearranged.
- The authors stated results in line 145. However, more details are needed so that the reader can understand the impact of this information.
Author Response
please see file attached

Reviewer 2 Report
The article entitled “Fast direct dihydrouracil quantitation in human saliva. Method 2 Development, Validation, and Application” by Campanella B etal., developed an HPLC-DAD based analytical method dihydrouracil in human saliva samples. The article lacks in several experimental data. I recommend authors to resubmit the manuscript after improving their manuscript.
Abstract: Should be revised. Background is insufficient, the importance and necessity of DHU measurement should be addressed. In results sections instead of comparing their results with previous reports (looks like discussion), the authors must provide their experimental results. For example, “DHU levels found in human saliva by HPLC-DAD confirm previous results obtained by LC-MS/MS” this sentence as no meaning in abstract section. Conclusions are missing.
Line 13: chemotherapics should be chemotherapeutics.
Line 48-49: Cite the reference appropriately.
Line 52-76: Avoid providing just 2-3 lines as one paragraph. Also, multiple units like ng/mL, μmol/L , μmol/mmol will confuse the readers. I suggest to unify it, better to use any one of the SI unit.
Line 98: delete concentration level.
Section 2.3: The study limits in terms of sample number as saliva is collected from only 8 volunteers at different time intervals. Authors must increase this number for better accuracy or limitation should be stated. Also, did authors found any changes in DHU with the time course? I think physiological concentration should be same, however external factors may influence. If they have the data they must provide and discuss it appropriately.
2.5 Method validation: This part is completely missing with experimental results. Authors must provide their experimental data on accuracy, precision, RSD as a new table. Also, n=3 is too low number for method validation studies. Calibration curve is missing.
Figure 1: Difficult to follow. There are multiple overlapping peaks with DHU. I suggest authors should provide the standard and in parallel sample chromatogram for better understanding. If this is a pure standard then what are these giant peaks in the chromatogram?
Line 156: “Table 1 reports a summary of the data from literature and the DHU 156 concentration values found in this work”, its unclear how did the authors calculate the concentration.
Table 1: does not show any concentration levels of DHU acquired in this study. If the concentrations are less than LOD what is the purpose this study? Results and discussion must be improved.
Author Response
please see file attached

Reviewer 3 Report
In this manuscript, Campanella et al., have developed a new method to determine dihydrouracil (DHU) levels in human saliva. Detection of DHU levels in biological fluids is important because any changes in the level of DHU could be an indicator of diseases ranging from autism to epilepsy. The authors have used liquid chromatography to monitor DHU levels in healthy patients. The authors used known conc. of DHU to establish the method and successfully demonstrated that DHU levels in healthy patients are below the detection limit. This suggests that DHU is not the primary metabolite of human saliva in healthy patients.
Overall, the data presented in this manuscript are of fair quality however, there are several aspects of this study that could help improve the major conclusions and the overall significance.
The method section requires more clarification, especially regarding how spiked samples were generated. A more detailed protocol for HPLC is required for a better understanding of the method.
The mention that linearity was calculated in the range of 0.001-2.5mmol L-1. It would be helpful to show that data.
The authors have tested saliva samples from eight healthy patients. Other reports have collected data from more patient samples. I believe eight is a very sample size and more patient samples need to be tested using this method. Additionally, saliva samples from patients undergoing chemotherapy should also be tested using this method.
Discussion: Could the author please comment on the limitation of this method.
Minor comments:
Few grammatical errors were observed in the manuscript.
Author Response
please see file attached

Reviewer 4 Report
The article Fast direct dihydrouracil quantitation in human saliva. Method, Development, Validation, and Application, by Beatrice Campanella, Tommaso Lomonaco, Edoardo Benedetti, Massimo Onor, Riccardo Nier and Emilia Bramanti, submitted for publication Biosensors, presents and propose a methodology for the quantification of the metabolic product dihydrouracil in saliva samples. Also, the validation of the methodology is presented.
The English language is fine but the article lacks the graphical representation of the results. In my opinion the validation should be included in the results section and supported with figures and tables. The same for sample analysis.
Considering all above mentioned, the article can be accepted for publication after a major revision.
Author Response
please see file attached

Round 2
Reviewer 2 Report
The authors significantly revised their manuscript based on the previous comments and added additional experimental data that required for an analytical paper. I recommend the manuscript for publication in IJERPH.
Minor comments:
Recheck for grammatical /typo errors thoroughly.
Figure 1D units should be corrected. Recommended adding the error bars in the curve as authors did n=3.
Since ethical number is not provided. Give the link of Helsinki that agree with below statement.
Institutional Review Board Statement: The study was conducted in accordance with the Declaration of Helsinki. Ethical review and approval are not applicable because all subjects were volunteers
Author Response
please see file attached

Reviewer 4 Report
After revision, the authors provided an improved version of their manuscript. In my opinion the article can be accepted for publication as it is.
Author Response
English language and style have been revised